environmental science/applied mathematics/complexity

machine learning, data science, deprivation

**Author for correspondence:**
Federico Botta
e-mail: f.botta@exeter.ac.uk

# Rapid indicators of deprivation using grocery shopping data

Adam Bannister[1] and Federico Botta[1,2]

[1]Department of Computer Science, University of Exeter, Exeter, UK
[2]The Alan Turing Institute, British Library, London, UK

FB, 0000-0002-5681-4535

Measuring socio-economic indicators is a crucial task for policy makers who need to develop and implement policies aimed at reducing inequalities and improving the quality of life. However, traditionally this is a time-consuming and expensive task, which therefore cannot be carried out with high temporal frequency. Here, we investigate whether secondary data generated from our grocery shopping habits can be used to generate rapid estimates of deprivation in the city of London in the UK. We show the existence of a relationship between our grocery shopping data and the deprivation of different areas in London, and how we can use grocery shopping data to generate quick estimates of deprivation, albeit with some limitations. Crucially, our estimates can be generated very rapidly with the data used in our analysis, thus opening up the opportunity of having early access to estimates of deprivation. Our findings provide further evidence that new data streams contain accurate information about our collective behaviour and the current state of our society.

## 1. Introduction

Good policies to tackle social and economic inequalities require accurate and timely knowledge of the state of society. However, many of the measures used by policy makers to influence their decisions are time-consuming and expensive to collect. They typically rely on large surveys, or on the gathering and processing of a wide range of secondary data from different parts of society. Recent years have witnessed a shift in this thanks to the unprecedented availability of vast volumes of data on our collective behaviour which are generated via our constant interactions with the Internet, mobile phones and many other large digital systems [1–3].

These new data streams have opened up the opportunity of studying several aspects of our collective behaviour and society, such as how people move [4–9], how diseases spread [10–13]

**Figure 1.** Comparison between deprivation and grocery shopping habits in London. We retrieve data on deprivation levels in 2015 for the city of London at the *Lower Superior Output Area (LSOA)* level as measured in the *Index of Multiple Deprivation (IMD)* by the *UK Ministry of Housing, Communities and Local Government*. The IMD is calculated by considering a variety of components which relate to deprivation in an area, such as income deprivation, crime levels and barriers to housing, and is used to rank each LSOA in the UK. (*a*) We depict the IMD ranks across the city of London for each LSOA. (*b*) We also retrieve data on grocery shopping habits in the supermarket chain *Tesco* in London in 2015 [35]. We aim to investigate whether grocery shopping data can be used to estimate the IMD. We depict the ranked weight of fibre in the average product for each LSOA in London. Visual inspection shows a strong similarity between areas of high deprivation and lower average fibre content. This provides initial evidence of a relationship between the IMD and aggregate grocery shopping data. Map tiles by *Stamen Design*, under CC BY 3.0. Data by *OpenStreetMap*, under ODbL.

and our social interactions [14–17]. The rapid increase in the urbanization of many countries, with cities quickly growing in size and population, has resulted in an expanding interest in using such novel data sources to study our cities [18–30].

Data derived from our interactions with the online and digital world have recently been used to design complementary measures of a range of socio-economic indicators used by policy makers [20,31–34]. In this paper, we investigate whether secondary data on our grocery shopping habits can be used to generate rapid estimates of the *English Indices of Multiple Deprivation* (IMD) in the city of London in the UK (figure 1). The IMD is an important measure for policy makers and local authorities, since it allows them to monitor deprivation levels across a city, region or even the whole country. The IMD can be used to help inform decisions on how to assign resources to tackle social and economic inequalities. While it is of great importance, the IMD is an expensive and time-consuming measure to calculate, and for this reason it is calculated and released only every 5 years in the UK. Crucially, data used to calculate the IMD for a given year often refers to up to 2 or 3 years before, thus introducing a significant lag in the data used to calculate it. The IMD is calculated from a range of different indicators linked to several aspects of deprivation (more details on this in the Data and methods section), such as economic and labour deprivation, but also health and living environment. Intuitively, we expect that some aspects of deprivation may be related to people's eating and shopping habits. Previous studies have shown how food deserts, which are areas with poor access to healthy food, can be inferred from food-related posts on Instagram [36], highlighting a link between food deprivation and what people eat. Here, we aim to investigate whether differences in grocery shopping habits between different areas of London are related to levels of deprivation. We rely on a large, publicly available dataset on grocery shopping habits in the city of London [35], and we use machine learning techniques to analyse the relationship between grocery purchases and the IMD. We construct a series of machine learning models aimed at investigating to what extent we can infer the IMD from grocery shopping data, and we also provide initial insight into what food items and nutritional values show the strongest connection with deprivation levels across London. Since data on grocery shopping habits could be made available to policy makers by private companies with little to no delay, at least in principle, we seek to investigate whether this data might enable us to generate rapid estimates of deprivation. Crucially, such estimates could be generated on an ongoing basis, for instance quarterly, providing a much more granular measure of the state of society.

# 2. Data and methods

We retrieve data on food items purchased at Tesco supermarkets in the Greater London area in 2015. Data are publicly available [35] and are derived from records of 420 millions of food items purchased by 1.6 million fidelity card owners. The data are available at the census area levels, and we use data at the Lower Super Output Area (LSOA) level. The dataset [35] is available at monthly granularity, but in our main analysis we consider yearly data for 2015, since that corresponds to the year for which the deprivation data described below is also available. However, we also test the sensitivity of our results to the amount of data used, and for this we will rely on the monthly data as well. For each LSOA, the dataset contains the average nutritional properties of the food purchased. Note that food purchases are not available at the individual level, so all nutritional values refer to an average over all food products purchased by residents of the LSOA. Products are divided among the following 17 categories: *beer, dairy, eggs, fats & oils, fish, fruit & veg, grains, red meat, poultry, readymade, sauces, soft drinks, spirits, sweets, tea & coffee, water* and *wine*. For each LSOA, the data contain the fraction of purchased products in each category, as well as the weight of different nutrients in the average product (measured in grams). The food nutrients included in the data are *carbs, sugar, fats, saturated fats, proteins* and *fibres*, and *alcohol*. Finally, a selection of derived measures from the grocery data is also included, such as measures of entropy, standard deviation and percentiles. In total, the data contains 188 different features constructed from food purchases. For a more complete and detailed description of the data, we refer the reader to the original study [35] where all the information on how the data was calculated and aggregated is available, as well as the exact description of all the features available in the dataset.

We want to compare the grocery shopping habits of residents in an LSOA with the deprivation level of that area, as measured by the *English Indices of Multiple Deprivation (IMD)*. The IMD is published in England by the Ministry of Housing, Communities & Local Government, and it provides a relative measure of deprivation of LSOAs in England. The IMD ranks each LSOA from most deprived to least deprived, and is further divided in seven domains of deprivation which constitute the overall index: *Income*; *Employment*; *Education*; *Health*; *Crime*; *Barriers to housing & services*; *Living environment*. Note that, while the IMD ranks all LSOAs in England, our analysis focuses only on London, so, prior to our analysis, we rank again the LSOAs in London so that we have a consistent ranking from 1 to 4833 (the total number of LSOAs in London). The IMD provides a relative measure of deprivation, meaning that it can only be used to compare different LSOA among each other, but it does not provide an estimate of how deprived each LSOA is. A range of indicators are used to calculate the IMD for each LSOA, such as the census, the number of claimants of *Jobseeker's Allowance*, recorded crime rates, entries to higher education, and several others [37]. Most indicators used to construct the 2015 IMD relate to the 2012/2013 tax year [37]. Here, we retrieve IMD data for 2015 [37], as it corresponds to the time period for which the grocery shopping data is also available.

We first carry out a non-parametric analysis using Kendall's rank correlation coefficient between the IMD data and each grocery shopping feature. Since this correlation analysis is carried out across all features, we adjust all resulting $p$-values using false discovery rate (fdr) correction to control the expected proportion of false positives [38]. To carry out the correction, we use the Python `statsmodels.stats.multitest.fdrcorrection` implementation [39]. This will give us a first indication of any link between deprivation levels across London and grocery shopping habits. Next, we train a random forest regression model to try and predict the IMD data from our grocery features. A random forest regression model is an ensemble of decision trees for regression. Each tree is trained with bagging and random sampling of the features at each candidate split of the learning process [40]. We decide to use random forests for our analysis as they can easily account for a large number of potentially correlated features, and can achieve good predictive performance. Additionally, we can use random forests to rank the importance of our features using out-of-bag errors. This is important in our analysis, as features which are strongly predictive of deprivation may be of interest to policy makers, since they could provide indicators of early signs of increasing, or decreasing, deprivation in an area. In the Results section, we present the results of our analysis when using a random forest with 50 decision trees, with each tree allowed a maximum depth of 15 nodes (note, however, that this is a limit on the maximum allowed depth only, but trees in the forest may also be smaller than this). The random forest is trained on 70% of the data (training set) and predictions are generated on the remaining 30% of data (test set). In the electronic supplementary material, figure S6, we present a sensitivity analysis of the random forest regression score for a range of values for the size of the forest

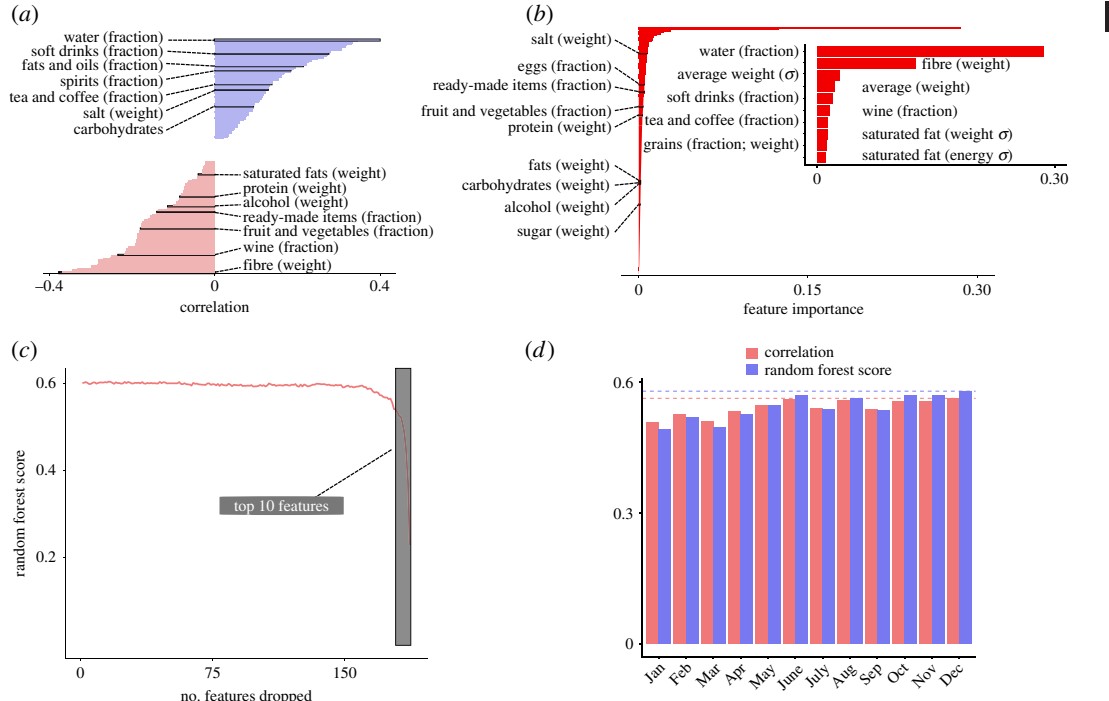

**Figure 2.** Investigating the relationship between deprivation and aggregated grocery shopping data. We investigate whether there is a relationship between the Index of Multiple Deprivation (IMD) and aggregate grocery shopping data in different areas of London. (*a*) We find that a variety of features extracted from the grocery shopping data exhibit a significant correlation with the IMD (Kendall's rank correlation coefficient; all depicted values are significant at the 5% level after adjusting using false discovery rate correction for multiple hypothesis testing). We highlight some features with a positive (light blue) or negative (light red) correlation. (*b*) We fit a random forest regression model on a sample of 70% of data. From the fitted random forest, we then extract the importance of each feature. Here, we depict the importance of each feature, and in the inset plot, we highlight the importance of the top 10 features in the model. When $\sigma$ appears in the feature name, this refers to the standard deviation of the corresponding underlying variable. For instance, *average weight* ($\sigma$) refers to the standard deviation of the weight of the average food product. (*c*) The feature importance of the fitted random forest suggests that a small subset of the features has most of the importance in the model. Here, we investigate what happens to the accuracy of the random forest as we start dropping features in increasing order of importance. Our findings show that removing most of the features does not significantly alter the score of the model. (*d*) We test the sensitivity of our model to the training data by constructing a series of random forest regression models based on monthly data. Each model is constructed using the monthly data available until the corresponding month of the year, e.g. the model from April uses data available from January until April. We depict both the score of the resulting random forest, as well as Kendall's correlation value between the true IMD data and the predicted values. Overall, we observe a good robustness of our model which performs well even with just one month of data. The best performance, highlighted by the horizontal dashed lines, is achieved by a model which uses the whole year of data.

and depth of each tree, providing a justification for the choice of parameters used here. Additionally, in the electronic supplementary material, we also present the results of our analysis when using adaptive boosting (AdaBoost) instead of random forests (electronic supplementary material, figures S4 and S5). The version of AdaBoost we use is given by an ensemble of decision trees which are trained sequentially, so that they can iteratively improve on the performance of the previous tree in the sequence. We find qualitatively similar results to random forests. Finally, our main analysis focuses on analysing the IMD ranks for the LSOAs in London, which is what is mostly of interest to policy makers. However, the IMD ranks are calculated from IMD scores which are calculated by the Ministry of Housing, Communities & Local Government [37]. Note that, while the IMD ranks are calculated from the scores, the two sets of analyses provide a slightly different approach. First, the published IMD scores are rounded versions of the scores used to rank the LSOAs. Second, the IMD scores are a truly continuous numerical variable, which may be better modelled by a regression model. Overall, we find qualitatively similar results using the IMD scores, and therefore we do not report the results of this analysis.

**Table 1.** Predictive power of different grocery categories. We investigate whether different categories of grocery items predict the Index of Multiple Deprivation with different levels of accuracy. We build three different models: one which only includes item types, such as meat and eggs; a second one which only includes nutrients, such as carbohydrates and fibres; and a third one which includes all variables. Each model consists of a random forest with 50 trees of maximum depth of 15 and is trained on 70% of the data, and the regression score is then calculated on the remaining 30%. We find a strong consistency in the results of our model, with similar scores across the three different models.

| model | random forest score |
|---|---|
| item types | 0.56 |
| nutrients | 0.57 |
| all data | 0.57 |

## 3. Results

Figure 2a depicts the results of our initial correlation analysis. Across most features, we find a significant correlation between the IMD and features derived from grocery shopping data at the LSOA level. As shown in the figure, we find that nearly half of the features exhibit a positive correlation, with the remaining exhibiting a negative correlation. The strongest correlated variables, respectively, positively and negatively, are the fraction of products in the category water and the weight of fibre in the average product purchased in each LSOA. Note that a positive correlation means, for instance, that more deprived areas tend to buy more water, and a negative correlation that more deprived areas tend to buy less fibre. Our findings suggest that there is a relationship between the relative level of deprivation of an area and the grocery shopping habits of people living there. Figure 2b presents the importance of all features as ranked by the random forest regression model, and confirms that the strongest correlated variables have the largest importance in the random forest.

 While having several features can often provide additional information and improve predictions, it is also important to balance this with the fact that grocery shopping data is not only privately held but most importantly ultimately refers to individuals. Thus, if the same result can be achieved with a smaller number of features, this reduces the amount of personal data needed to construct features which are useful to estimate the IMD. This is beneficial in terms of a reduced need of accessing personal data, and may be more palatable to private companies, which would only need to share a small number of aggregated data. We therefore test how the random forest model performs if we start removing features for training. In particular, we remove features one by one in increasing order of importance, starting from the least important. Figure 2c depicts the random forest score, measured by the coefficient of determination $R^2$, as a function of the number of features dropped from the model. We find that the model score does not change significantly for a very large number of features dropped, and only starts decreasing once the last, most important, features are removed. This provides evidence that a random forest model can achieve a good $R^2$ score with only a small number of features derived from grocery shopping data. Finally, it is also of interest to know if different categories of grocery items are more predictive of deprivation. This may provide valuable information to policy makers who can gain a better insight into possible early indicators of deprivation, as well as to better understand links between food habits, health and socio-economic outcomes. Therefore, we construct three different versions of our random forest: first, we use all grocery shopping data as above; second, we only use data from the nutrients contained in the purchased products, such as carbohydrates and fibres; lastly, we use item types, such as meat or eggs, as our only predictors. Table 1 presents the results of this analysis. Overall, we find a very strong consistency of our results in the three models.

 So far, our analysis has used yearly grocery shopping data. However, data on grocery shopping habits could, in principle, be made available at a much higher temporal granularity. Therefore, we test the sensitivity of our results to how much data we use. We build a series of random forest models using monthly data, and we sequentially include all months from the start of the year to the current month under consideration, starting from January 2015. For instance, the first model only uses data for January 2015; a model in April, instead, uses all data available from January until April 2015. Figure 2d depicts the results of this analysis both in terms of the random forest regression score, as well as Kendall's correlation value between the true and predicted IMD. Overall, we find that our model can produce good results even with just one month of data. We also note that the best results,

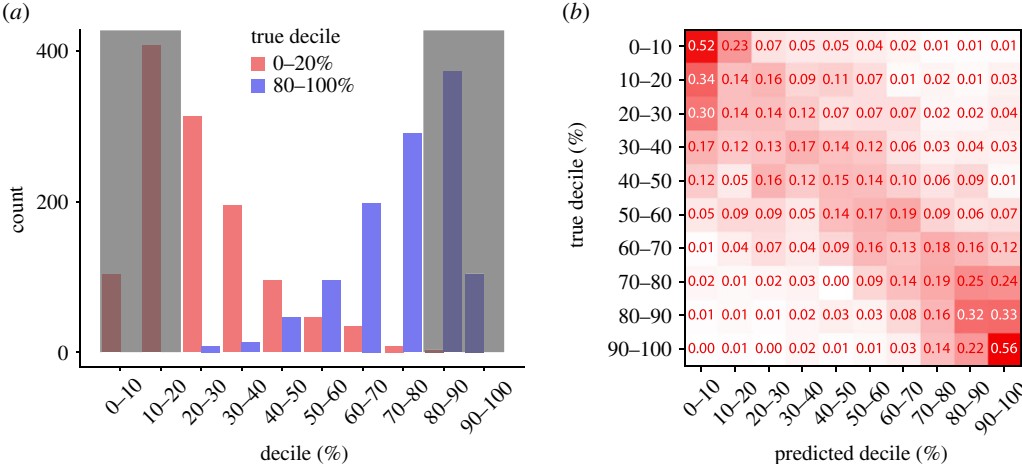

**Figure 3.** Estimating relative deprivation using grocery shopping data. (*a*) We use a random forest to predict the areas in London not used in training the model. Here, we present the results only for the most (red) and least (blue) deprived areas of London. On the *x*-axis, we show the deciles based on the predicted IMD values. This means that, for instance, an LSOA whose true IMD is in the bottom 10% decile, which is incorrectly predicted to fall in the 10–20% decile, would contribute to the red 10–20% bar in this figure. Broadly speaking, we find that our model is able to discriminate between the areas, but also that it misclassifies a number of them. (*b*) We investigate whether we can generate estimates of the decile of relative deprivation to which each LSOA belongs to. We train a random forest classifier using grocery shopping data as predictor variables, and the decile of deprivation of each LSOA as predicted variable. We use 70% of the data to train the model, and we test the results on the remaining 30%. Here, we depict the confusion matrix of the predictions, where each row is normalized. We find that the classifier is able to differentiate well between the most and least deprived LSOAs, with minimal overlap between the two extremes. LSOAs in the middle range of deciles are predicted less accurately, as it is probably harder to differentiate among them.

both in terms of regression score and correlation, are obtained when using a whole year of data, but the difference is relatively small. This result indicates that it may not be necessary to use data for a whole year, thus reducing the need of accessing personal data.

Finally, we investigate to what extent our random forest model is able to generate predictions of the IMD which reflect the relative deprivation of different LSOAs in London. We train a random forest on 70% of the data and predict the remaining 30%, and we repeat this procedure four times. This means that at each iteration we test our model on 1450 LSOAs (corresponding to 30% of the 4833 LSOAs of London), which over four repetitions gives us a total sample size of 5800. Since relative levels of deprivation, rather than the actual scores, are the key information contained in the IMD, a good measure of goodness of our predictions can be calculated using Kendall's correlation coefficient, which is a rank-based correlation measure, as also previously suggested [33]. In our analysis, we find a correlation coefficient of 0.57 ($N = 5800$; $p < 0.001$; Kendall's $\tau$ correlation coefficient). This result is in very good agreement with other estimates of the IMD generated from new forms of data [33], and provides further evidence that rapid estimates of the IMD can be obtained from secondary data sources about human behaviour. To further analyse the accuracy, we select all LSOAs in the test set (of each of the four iterations discussed above) which are in the top 20% and bottom 20% of deprivation, and see whether they are predicted to be in the correct deciles by our random forest model. Figure 3*a* depicts the results of this analysis. In red, we represent the LSOAs which are in the top 20%, and in blue the bottom 20%. On the *x*-axis, we show the predicted decile. For instance, an LSOA which is in the bottom 10% decile of deprivation, and which is incorrectly predicted to be in the 10–20% decile by our model, would contribute to the red bar in the 10–20% decile on the *x*-axis. Broadly speaking, the random forest is able to separate least and most deprived areas to a good extent. However, this result also shows the limitations of trying to predict a complex socio-economic indicator such as the IMD-based solely on aggregated grocery shopping data.

A different, and complementary, approach is to focus on directly predicting the IMD decile of each LSOA rather than its actual rank. We test this approach by fitting a random forest classifier, where in this case the random forest is trained to classify each LSOA in the corresponding decile class. Figure 3*b* depicts the results of this classification as a confusion matrix, where each row has been normalized. We find that the random forest is able to discriminate very well between least and most

deprived LSOAs, as demonstrated by the high values in the top left and bottom right corners of the confusion matrix. However, LSOAs corresponding to deciles in the middle of the range are harder to classify and the random forest does not differentiate them as well, a result already observed previously [33]. Overall, this shows that it is possible to classify LSOAs which are in the top and bottom deciles with a very good accuracy. This supports our hypothesis that estimates of deprivation generated using secondary data could be helpful for policy makers in identifying the areas which are either relatively least or most deprived. While this does not provide accurate estimates for all LSOAs, it could still be a very useful tool in the hands of policy makers who are interested in understanding if their policies have helped the most deprived areas.

Lastly, we want to investigate to what extent the results found in our analysis could be due to chance alone. We test this by randomly reshuffling the grocery shopping data, so that each value is assigned to a randomly selected LSOA, and we then calculate Kendall's correlation coefficient between each reshuffled variable and the IMD. In practice, this is equivalent to spatially reshuffling the data. We repeat the random reshuffle 1000 times for each variable, and we adjust all resulting correlation values using false discovery rate correction. For the majority of features (156 out of 169), we find no significant correlation values after reshuffling. For the remaining 13 variables, we find that less than 0.01% of the correlation values are significant after reshuffling. This provides a strong evidence for the robustness of our results.

## 4. Discussion

Our analysis has shown the existence of a relationship between what food people purchase in a supermarket and the level of deprivation of LSOAs in London. This finding suggests that policy makers interested in measuring the current state of society may be able to use such data to generate rapid indicators of levels of deprivation. Additionally, our analysis has also shown that only a small subset of grocery shopping features is needed to construct a predictive model. This result is important in the context of data sharing, since it indicates that policy makers may be able to generate indicators of deprivation by using only a small number of features, thus reducing the need of having access to large amounts of personal data. This may also be attractive to private supermarket companies, who may have concerns in sharing customers data. Our findings also demonstrate that we can use grocery shopping data to accurately classify areas which are least and most deprived, giving policy makers early access to indicators of which areas may be suffering from high levels of deprivation or may be thriving.

We also find that our analysis produces results in agreement with the findings of previous research on nutritional intakes between different socio-economic groups. Existing studies show that those who are more health-conscious, and therefore tend to buy healthier food, are said to be less price sensitive [41]. Evidence shows that people in lower socio-economic classes tend to have a less healthy diet throughout their entire life [42], while people in higher socio-economic groups tend to have better and easier access to healthy foods [29]. The weight of fibre in the average product, and the amount of fruit and vegetables, were strongly correlated towards less deprived areas, supported by previous research which found that healthier areas buy more fibre, as well as less sugar. Interestingly, our analysis has shown a strong link between purchase of water and deprivation, with more deprived areas tending to buy more water, a result which needs further exploring in future work. It is also important to emphasize that our analysis provides no insight into the direction of the links found between deprivation and grocery shopping habits, therefore no inference about causality can be made from our results.

Our analysis opens up the potential opportunity of generating rapid estimates of the levels of deprivation using secondary data alone. Data on food purchases is collected by most supermarket companies in order to understand their customer behaviour. Our findings suggest that such secondary data could be used by policy makers to also generate estimates of deprivation, particularly to identify areas which are at the ends of the IMD distribution. Crucially, such data could be made available at a much more rapid pace than traditional sources which feed into the IMD, thus opening up the opportunity of generating more frequent estimates of deprivation. This would be of crucial importance to have rapid measurements of the state of our society, enabling policy makers to assess more quickly the impact of policies designed to tackle deprivation. Traditional measures of deprivation would still be required for regular calibration of the rapid indicators, but our analysis suggests that they could be complemented by alternative measures generated with the analysis of new data streams.

However, it is also important to mention that our analysis suffers from some limitations. Firstly, the grocery shopping data available for analysis, while derived from millions of transactions, only refers to one specific supermarket chain. People with different socio-economic status may shop in different supermarkets. For instance, people suffering from extreme deprivation may only be able to shop in discount supermarket, or may have to rely on food banks. Similarly, wealthy people may shop in small, high-quality independent shops, or may even purchase less food because they can afford to eat out more regularly. Additionally, the geographical distribution of Tesco shops in London is not uniform [35]. This clearly affects the shopping habits of people living in different areas of London. Furthermore, different branches of Tesco shops may have slightly different ranges of products for sale, thus affecting what people in different areas can buy. Data on this are currently unavailable, but this could be an interesting avenue for further research. Additionally, the IMD is composed of a variety of indicators that take into account a wide range of factors affecting deprivation, which include, but are not limited to what people buy and eat. And lastly, while grocery shopping data was made publicly available in this case, this may not always be possible or feasible. However, we anticipate that further, related secondary data sources could give analogous results [33].

Our findings provide further evidence that rapidly available secondary data on our collective behaviour can provide fast insights into the current state of our society. Fast, up-to-date data can help reduce delays in designing and understanding policies to improve our society.

Data accessibility. Data are publicly available at [35,43] and via the UK Ministry of Housing, Communities & Local Government at https://www.gov.uk/government/statistics/english-indices-of-deprivation-2015. All code needed to reproduce the analysis is available at 10.5281/zenodo.5139534.
Authors' contributions. F.B. designed the study concept. F.B. and A.B. carried out the statistical analyses. F.B. wrote the manuscript. All authors gave final approval for publication.
Competing interests. We declare we have no competing interests.
Funding. The authors received no funding for this manuscript.
Acknowledgements. We thank Luca Maria Aiello, Daniele Quercia, Rossano Schifanella and Lucia Del Prete for making the data publicly available [35].

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
