## [Peer Review File · Royal Society Open Science]

Review History

RSOS-211069.R0 (Original submission)

Review form: Reviewer 1

Is the manuscript scientifically sound in its present form?

No

Are the interpretations and conclusions justified by the results?

No

Is the language acceptable?

Yes

Do you have any ethical concerns with this paper?

No

Have you any concerns about statistical analyses in this paper?

Yes

Recommendation?

Major revision is needed (please make suggestions in comments)

Comments to the Author(s)

The paper uses data from grocery shopping sales to estimate deprivation at the level of small neighborhoods in London (LSOA). The idea is quite simple, yet very much worth exploring: the ability of estimating economic distress in near real time with "alternative" sources of data makes a lot of sense, as the authors explain well in the introduction. The paper is well-motivated and well-written, and I would very much like to see a contribution of this kind published. I would therefore use the opportunity of this review to give the authors some suggestions to make the contribution as impactful as possible and to fix some of its issues.

There are indeed some major issues that need to be addressed.

=== 1. There's an issue with the analysis (rank vs. score?)

The results do not seem to be correct. A warning bell rang in my brain when I saw the results in Figure 1, which shows that deprived areas are those that consume more fibres. This did not sound right as intuition would suggest that high-income (/high-education) areas would consume healthier food, and thus more fibre. I checked the paper "Large-scale and high-resolution analysis of food purchases and health outcomes", published from the same authors who made the Tesco dataset available. In Figure 3, I found a choropleth showing the opposite result from what's presented in this submission: low income areas consume less fibre. So I decided to check myself. I downloaded the Tesco open data [1] and the IMD data from 2015. I found that fibre consumption correlates positively (R close to 0.5) with the IMD rank. Higher rank corresponds to richer areas (e.g., City of London 001A, E09000001, has a score of 6.2 and a rank of 29199). I also checked manually some of the areas:

LSOA with supposedly high fibre consumption (according to this paper):

E01001409 -> is actually a low-fibre consumption area, ranked 3873 (e.g., postcode=EN1 4UP, latlon = 51.661541,-0.0619929)

E01003508 -> is actually a low-fibre consumption area, ranked 4449 (e.g., postcode=E16 1QB, latlon= 51.515732,0.018261)

LSOA with supposedly low fibre consumption (according to this paper):

E01001335 -> is actually a high-fibre consumption area, ranked 533 (e.g., postcode=UB2 4LG, latlon=51.495106,-0.366001)

I think that there might be two problems here:

- a mixup between IMD rank and IMD scores (one is the opposite of the other)
- an incorrect interpretation of the values from the Tesco dataset

This casts a doubt on the results presented. I am quite confident that the magnitude of results reported is correct, I am not sure about the sign though -- and that's a major issue because it would lead to a complete different interpretation of the results.

=== 2. The key grocery elements of deprivation

There is very little discussion about the types of products that are more predictive of deprivation. I believe that the top predictors should be listed more exhaustively and discussed. I would also strongly suggest to perform three different predictions, one using just the nutrients (fibres, carbs..), one using just the item types (meat, sweets..), and one using all of them (the one that it is currently presented). It would be interesting to see which family of features is most predictive. Then, I would expect an expanded discussion or result section that contains some hypotheses (or even some speculations) of why some factors might be more predictive -- fibres is an easy example, but maybe there are more interesting associations to be discussed. This addition would

add a considerable value to the contribution, in my opinion. If there are strange associations (i.e., not explained by intuition or previous work) it would be worth pointing them out.

=== 3. The temporal aspect

I totally see why the authors decided to go for the full grocery dataset. Yet, a relevant piece of analysis that ties directly into their research question is how little data one can use to make a good IMD assessment from grocery data. At the moment, the authors are using one full year of data, but a sensitivity analysis that uses 1,2,3,6 months of data could reveal that smaller amounts of data could suffice for such an estimation.

=== Minor comments:

- Using decision trees with depth of 150 is definitely quite an overkill. This is also made quite clear by the analysis in the supplementary materials, which shows that the depth of the tree does not influence the results. I believe that a depth of 5 would do just as fine. I would strongly recommend to revise the presentation of the main results using trees of limited depth.

- It is not necessary to repeat the analysis for IMD ranks and scores (in the supplementary materials). Those are completely symmetrical (so the results should not be "similar" but identical, just with a flipped sign). So just be very clear and explicit about which one you are using and about its meaning.

- Which procedure did you use to implement the false discovery rate correction? It would be best to report it clearly (even just mentioning the name of the software package and function used)

- There are no spaces before the in-text citations. For example, "available[36]" should be "available [36]". You can use `text~\cite{xyz}` to have a space and keep the citation close to the previous word.

Notes:

[1] https://figshare.com/articles/dataset/Area-level_grocery_purchases/7796666?backTo=/collections/Tesco_Grocery_1_0/4769354, file year_lsoa_grocery, column "fibre"

Review form: Reviewer 2

Is the manuscript scientifically sound in its present form?

Yes

Are the interpretations and conclusions justified by the results?

Yes

Is the language acceptable?

Yes

Do you have any ethical concerns with this paper?

No

Have you any concerns about statistical analyses in this paper?

No

Recommendation?

Major revision is needed (please make suggestions in comments)

Comments to the Author(s)

This is a very interesting paper that looks at a very novel dataset in a novel setting. I very much enjoyed reading it. I think this is a promising paper, but needs revisions for a good publication.

Data and methods descriptions, and results need substantial improvements to clarify:

- Page 4, Lines 50-51: Is there a full list of product categories and nutrients, along with how they are measured e.g. fractions in terms of items, grams, purchase value? Not clear from the text what the 188 features refer to. How come 19 of these features have a constant value? While there is a reference to another paper – I think the present paper should include a better and clearer definition here to make sure the reader can better understand the data used for this analysis.

- If the IMD rankings are available for the whole UK, this means the rankings will take into account not only London LSOAs but also other LSOAs around the country. When being used for this analysis, do the rankings refer only to London, or UK. How might this affect results?

- Page 5, paragraph starting with Line 21 needs clarification - in its current format, it is very hard to understand what has been done for this analysis and why. Is there a reason
Page 6, Lines 24-27 suggest that five-fold cross validation is done. Not clear how this is done with 70%-30% splits. This might be a typo, but then earlier Page 5, Lines 33-35 suggests that no cross validation is done, but only the 70-30% splits are done. This needs to be clarified. In the current setting, cross-validation is preferred. Results also need to reflect the differences in findings in different folds.

How is a single accuracy measure computed - on the full London data? Also Not sure where $N=5,800$ comes from. This seems higher than the number of LSOA's in London.

Not clear what positive vs. negative correlation mean for Figure 2A (also in the text). The more the fibre, the better off the neighbourhood? Needs clarification what negative and positive means in the setting. Also, the paper will benefit from a discussion on whether these findings are intuitive or not. The more money(or items) as a fraction bought from Tesco, the worse off the neighbourhood? Do the authors find these results intuitive, and why?

Very hard to understand Figure 2D - the main text and caption are both confusing. Needs clarification. Are these all LSOAs in London, or only the 30%

Do we know if/how Tesco's product ranges offered in different stores make a difference?

How are findings compare to existing literature on food choices by different income groups. There is a rich literature that focuses specifically on diets of different socioeconomic groups with a specific interest in poor neighbourhoods. The manuscript will benefit from clarifying the links to this literature base.

Not clear to me if Tesco data is also available for later years, and how it was accessed. It is publicly available - but I was not sure if the year 2015 was specifically selected to match the IMD data. If available from Tesco, data from the year 2019 will be key to understand if these datasets can be used for predictions. specifically, food preferences and contents change over time - we do not know how it will effect predictions. If this method was used in practice, one would need to train models from one year data and make predictions in following years. Not clear to me how the changing preferences and food content (from the supply side) will affect predictive

capabilities. The paper should address such concerns. If 2019 data is available, this will be the ideal test case. Can a model trained on 2015 data make good predictions on 2019 data. As we already have 2019 IMD data, this can be tested.

It will be interesting to see how Tesco stores in addition to other retailers that cater to lower or higher income groups (Waitrose vs. Iceland) are distributed across space. Could that information be somehow used in relation to IMD. I would expect to observe a high correlation between existence of certain chains in certain LSOAs based on their target populations.

Last point on motivation. The authors motivate their work with an emphasis on prediction. Yet the train-test strategy (both over space, and over time) does not seem to be well-designed to fit the motivation. This needs clarification. Descriptive results on their own, are very interesting. The focus on predictive capabilities, however, needs a stronger argument and better justification for the train/test split strategies.

Decision letter (RSOS-211069.R0)

Dear Dr Botta

The Editors assigned to your paper RSOS-211069 "Rapid indicators of deprivation using grocery shopping data" have now received comments from reviewers and would like you to revise the paper in accordance with the reviewer comments and any comments from the Editors. Please note this decision does not guarantee eventual acceptance.

Please submit your revised manuscript and required files (see below) no later than 21 days from today's (ie 11-Oct-2021) date. Note: the ScholarOne system will 'lock' if submission of the revision is attempted 21 or more days after the deadline. If you do not think you will be able to meet this deadline please contact the editorial office immediately.

on behalf of Marta Kwiatkowska (Subject Editor)
 openscience@royalsociety.org

Associate Editor Comments to Author:

Comments to the Author:

The reviewers have offered substantial commentary and queries on your manuscript - it is clear that a lot of work has gone into both the paper and the reviewers' reports, and we hope the latter prove useful in guiding your revision. Good luck!

Reviewer comments to Author:

Reviewer: 1

Comments to the Author(s)

The paper uses data from grocery shopping sales to estimate deprivation at the level of small neighborhoods in London (LSOA). The idea is quite simple, yet very much worth exploring: the ability of estimating economic distress in near real time with "alternative" sources of data makes a lot of sense, as the authors explain well in the introduction. The paper is well-motivated and well-written, and I would very much like to see a contribution of this kind published. I would therefore use the opportunity of this review to give the authors some suggestions to make the contribution as impactful as possible and to fix some of its issues.

There are indeed some major issues that need to be addressed.

=== 1. There's an issue with the analysis (rank vs. score?)

The results do not seem to be correct. A warning bell rang in my brain when I saw the results in Figure 1, which shows that deprived areas are those that consume more fibres. This did not sound right as intuition would suggest that high-income (/high-education) areas would consume healthier food, and thus more fibre. I checked the paper "Large-scale and high-resolution analysis of food purchases and health outcomes", published from the same authors who made the Tesco dataset available. In Figure 3, I found a choropleth showing the opposite result from what's presented in this submission: low income areas consume less fibre. So I decided to check myself. I downloaded the Tesco open data [1] and the IMD data from 2015. I found that fibre consumption correlates positively (R close to 0.5) with the IMD rank. Higher rank corresponds to richer areas (e.g., City of London 001A, E09000001, has a score of 6.2 and a rank of 29199). I also checked manually some of the areas:

LSOA with supposedly high fibre consumption (according to this paper):

E01001409 -> is actually a low-fibre consumption area, ranked 3873 (e.g., postcode=EN1 4UP
 latlon = 51.661541,-0.0619929)

E01003508 -> is actually a low-fibre consumption area, ranked 4449 (e.g., postcode=E16 1QB,
 latlon= 51.515732,0.018261)

LSOA with supposedly low fibre consumption (according to this paper):

E01001335 -> is actually a high-fibre consumption area, ranked 533 (e.g., postcode=UB2 4LG,
 latlon=51.495106,-0.366001)

I think that there might be two problems here:

- a mixup between IMD rank and IMD scores (one is the opposite of the other)
- an incorrect interpretation of the values from the Tesco dataset

This casts a doubt on the results presented. I am quite confident that the magnitude of results reported is correct, I am not sure about the sign though -- and that's a major issue because it would lead to a complete different interpretation of the results.

=== 2. The key grocery elements of deprivation

There is very little discussion about the types of products that are more predictive of deprivation. I believe that the top predictors should be listed more exhaustively and discussed. I would also strongly suggest to perform three different predictions, one using just the nutrients (fibres, carbs..), one using just the item types (meat, sweets..), and one using all of them (the one that it is currently presented). It would be interesting to see which family of features is most predictive. Then, I would expect an expanded discussion or result section that contains some hypotheses (or even some speculations) of why some factors might be more predictive -- fibres is an easy example, but maybe there are more interesting associations to be discussed. This addition would add a considerable value to the contribution, in my opinion. If there are strange associations (i.e., not explained by intuition or previous work) it would be worth pointing them out.

=== 3. The temporal aspect

I totally see why the authors decided to go for the full grocery dataset. Yet, a relevant piece of analysis that ties directly into their research question is how little data one can use to make a good IMD assessment from grocery data. At the moment, the authors are using one full year of data, but a sensitivity analysis that uses 1,2,3,6 months of data could reveal that smaller amounts of data could suffice for such an estimation.

=== Minor comments:

- Using decision trees with depth of 150 is definitely quite an overkill. This is also made quite clear by the analysis in the supplementary materials, which shows that the depth of the tree does not influence the results. I believe that a depth of 5 would do just as fine. I would strongly recommend to revise the presentation of the main results using trees of limited depth.

- It is not necessary to repeat the analysis for IMD ranks and scores (in the supplementary materials). Those are completely symmetrical (so the results should not be "similar" but identical, just with a flipped sign). So just be very clear and explicit about which one you are using and about its meaning.

- Which procedure did you use to implement the false discovery rate correction? It would be best to report it clearly (even just mentioning the name of the software package and function used)

- There are no spaces before the in-text citations. For example, "available[36]" should be "available [36]". You can use `text~\cite{xyz}` to have a space and keep the citation close to the previous word.

Notes:

[1] https://figshare.com/articles/dataset/Area-level_grocery_purchases/7796666?backTo=/collections/Tesco_Grocery_1_0/4769354, file year_lsoa_grocery, column "fibre"

Reviewer: 2

Comments to the Author(s)

This is a very interesting paper that looks at a very novel dataset in a novel setting. I very much enjoyed reading it. I think this is a promising paper, but needs revisions for a good publication.

Data and methods descriptions, and results need substantial improvements to clarify:

- Page 4, Lines 50-51: Is there a full list of product categories and nutrients, along with how they are measured e.g. fractions in terms of items, grams, purchase value? Not clear from the text what the 188 features refer to. How come 19 of these features have a constant value? While there is a reference to another paper – I think the present paper should include a better and clearer definition here to make sure the reader can better understand the data used for this analysis.

- If the IMD rankings are available for the whole UK, this means the rankings will take into account not only London LSOAs but also other LSOAs around the country. When being used for this analysis, do the rankings refer only to London, or UK. How might this affect results?

- Page 5, paragraph starting with Line 21 needs clarification - in its current format, it is very hard to understand what has been done for this analysis and why. Is there a reason
Page 6, Lines 24-27 suggest that five-fold cross validation is done. Not clear how this is done with 70%-30% splits. This might be a typo, but then earlier Page 5, Lines 33-35 suggests that no cross validation is done, but only the 70-30% splits are done. This needs to be clarified. In the current setting, cross-validation is preferred. Results also need to reflect the differences in findings in different folds.

How is a single accuracy measure computed - on the full London data? Also Not sure where $N=5,800$ comes from. This seems higher than the number of LSOA's in London.

Not clear what positive vs. negative correlation mean for Figure 2A (also in the text). The more the fibre, the better off the neighbourhood? Needs clarification what negative and positive means in the setting. Also, the paper will benefit from a discussion on whether these findings are intuitive or not. The more money(or items) as a fraction bought from Tesco, the worse off the neighbourhood? Do the authors find these results intuitive, and why?

Very hard to understand Figure 2D - the main text and caption are both confusing. Needs clarification. Are these all LSOAs in London, or only the 30%

Do we know if/how Tesco's product ranges offered in different stores make a difference?

How are findings compare to existing literature on food choices by different income groups. There is a rich literature that focuses specifically on diets of different socioeconomic groups with a specific interest in poor neighbourhoods. The manuscript will benefit from clarifying the links to this literature base.

Not clear to me if Tesco data is also available for later years, and how it was accessed. It is publicly available - but I was not sure if the year 2015 was specifically selected to match the IMD data. If available from Tesco, data from the year 2019 will be key to understand if these datasets can be used for predictions. specifically, food preferences and contents change over time - we do not know how it will effect predictions. If this method was used in practice, one would need to train models from one year data and make predictions in following years. Not clear to me how the changing preferences and food content (from the supply side) will affect predictive capabilities. The paper should address such concerns. If 2019 data is available, this will be the ideal test case. Can a model trained on 2015 data make good predictions on 2019 data. As we already have 2019 IMD data, this can be tested.

It will be interesting to see how Tesco stores in addition to other retailers that cater to lower or higher income groups (Waitrose vs. Iceland) are distributed across space. Could that information be somehow used in relation to IMD. I would expect to observe a high correlation between existence of certain chains in certain LSOAs based on their target populations.

Last point on motivation. The authors motivate their work with an emphasis on prediction. Yet the train-test strategy (both over space, and over time) does not seem to be well-designed to fit the motivation. This needs clarification. Descriptive results on their own, are very interesting. The focus on predictive capabilities, however, needs a stronger argument and better justification for the train/test split strategies.

===PREPARING YOUR MANUSCRIPT===

===PREPARING YOUR REVISION IN SCHOLARONE===

Author's Response to Decision Letter for (RSOS-211069.R0)

See Appendix A.

RSOS-211069.R1 (Revision)

Review form: Reviewer 1

Is the manuscript scientifically sound in its present form?

Yes

Are the interpretations and conclusions justified by the results?

Yes

Is the language acceptable?

Yes

Do you have any ethical concerns with this paper?

No

Have you any concerns about statistical analyses in this paper?

No

Recommendation?

Accept as is

Comments to the Author(s)

I am pleased to see that the authors took all my comments very seriously and produced a version of the manuscript that is much improved. I am particularly glad that they were able to fix the mistake in Figure 1. I note the fact that, because of rounding, ranks and scores might be slightly different (I had never thought of that). The temporal analysis is interesting as it shows a peak at 6months but a quite decent performance already at 1month.

I am happy with the revised version!

Review form: Reviewer 2

Is the manuscript scientifically sound in its present form?

Yes

Are the interpretations and conclusions justified by the results?

Yes

Is the language acceptable?

Yes

Do you have any ethical concerns with this paper?

No

Have you any concerns about statistical analyses in this paper?

No

Recommendation?

Accept as is

Comments to the Author(s)

All my comments and questions were addressed in detail.

Decision letter (RSOS-211069.R1)

Dear Dr Botta,

It is a pleasure to accept your manuscript entitled "Rapid indicators of deprivation using grocery shopping data" in its current form for publication in Royal Society Open Science. The comments of the reviewer(s) who reviewed your manuscript are included at the foot of this letter.

The proof of your paper will be available for review using the Royal Society online proofing system and you will receive details of how to access this in the near future from our production office (opencscience_proofs@royalsociety.org). We aim to maintain rapid times to publication after acceptance of your manuscript and we would ask you to please contact both the production office and editorial office if you are likely to be away from e-mail contact to minimise delays to publication. If you are going to be away, please nominate a co-author (if available) to manage the proofing process, and ensure they are copied into your email to the journal.

Kind regards,
Royal Society Open Science Editorial Office
Royal Society Open Science
opencscience@royalsociety.org

on behalf of Prof Marta Kwiatkowska (Subject Editor)
opencscience@royalsociety.org

Reviewer comments to Author:

Reviewer: 1

Comments to the Author(s)

I am pleased to see that the authors took all my comments very seriously and produced a version of the manuscript that is much improved. I am particularly glad that they were able to fix the mistake in Figure 1. I note the fact that, because of rounding, ranks and scores might be slightly different (I had never thought of that). The temporal analysis is interesting as it shows a peak at 6months but a quite decent performance already at 1month.

I am happy with the revised version!

Reviewer: 2

Comments to the Author(s)

All my comments and questions were addressed in detail.

Appendix A

Federico Botta
Lecturer in Data Science

College of Engineering, Mathematics and
Physical Sciences
Harrison Building
Streatham Campus
University of Exeter
North Park Road
UK, EX4 4QF

+44 (0)1392 722220
f.botta@exeter.ac.uk

25 October 2021

Submission of “Rapid indicators of deprivation using grocery shopping data”

Dear Marta Kwiatkowska (Subject Editor),

We were extremely pleased to received your decision letter on 11th October 2021 with the positive feedback from the two reviewers. We were glad to hear that the reviewers agreed that the results were interesting, supported by the analysis and well written.

The reviewers’ feedback was indeed very useful and valuable, and has allowed us to further improve the quality of our manuscript. We would like to sincerely thank them since their thoughts and ideas have been of great interest, and have improved our work.

We include below here the comments of the reviewers, as well as our responses to each of their points. Additionally, we also include a PDF copy of our manuscript which highlights the changes we did as a result of the feedback from the reviewers.

We greatly appreciate the time taken by you and the reviewers to provide helpful and constructive feedback, which we are confident we have addressed in our revised manuscript. We hope that our manuscript is now ready to be published in *Royal Society Open Science*.

Thank you once again for your kind help and consideration in this matter.

Sincerely,

Federico Botta, and Adam Bannister

Reviewer 1

The paper uses data from grocery shopping sales to estimate deprivation at the level of small neighborhoods in London (LSOA). The idea is quite simple, yet very much worth exploring: the ability of estimating economic distress in near real time with "alternative" sources of data makes a lot of sense, as the authors explain well in the introduction. The paper is well-motivated and well-written, and I would very much like to see a contribution of this kind published. I would therefore use the opportunity of this review to give the authors some suggestions to make the contribution as impactful as possible and to fix some of its issues.

We thank the reviewer for their positive general considerations on our manuscript. Their suggestions have helped greatly improve the manuscript, in particular with respect to some inaccuracies that were present in our initial submission. We are grateful to the reviewer for taking the time to consider those and suggesting ways of improving the submission, which we now feel is in a much better shape and ready for publication.

There are indeed some major issues that need to be addressed.

We agree that the reviewer has highlighted some important areas to improve, and we have tried to do so extensively, both by improving the analysis as well as the presentation. We respond to each comment in more detail below.

=== 1. There's an issue with the analysis (rank vs. score?)

The results do not seem to be correct. A warning bell rang in my brain when I saw the results in Figure 1, which shows that deprived areas are those that consume more fibres. This did not sound right as intuition would suggest that high-income (/high-education) areas would consume healthier food, and thus more fibre. I checked the paper "Large-scale and high-resolution analysis of food purchases and health outcomes", published from the same authors who made the Tesco dataset available. In Figure 3, I found a choropleth showing the opposite result from what's presented in this submission: low income areas consume less fibre. So I decided to check myself. I downloaded the Tesco open data [1] and the IMD data from 2015. I found that fibre consumption correlates positively (R close to 0.5) with the IMD rank. Higher rank corresponds to richer areas (e.g., City of London 001A, E09000001, has a score of 6.2 and a rank of 29199). I also checked manually some of the areas:

LSOA with supposedly high fibre consumption (according to this paper):

E01001409 -> is actually a low-fibre consumption area, ranked 3873 (e.g., postcode=EN1 4UP latlon = 51.661541,-0.0619929)

E01003508 -> is actually a low-fibre consumption area, ranked 4449 (e.g., postcode=E16 1QB, latlon= 51.515732,0.018261)

LSOA with supposedly low fibre consumption (according to this paper):

E01001335 -> is actually a high-fibre consumption area, ranked 533 (e.g., postcode=UB2 4LG, latlon=51.495106,-0.366001)

I think that there might be two problems here:

- a mixup between IMD rank and IMD scores (one is the opposite of the other)**
- an incorrect interpretation of the values from the Tesco dataset**

This casts a doubt on the results presented. I am quite confident that the magnitude of results reported is correct, I am not sure about the sign though -- and that's a major issue because it would lead to a complete different interpretation of the results.

We would like to thank the reviewer for their careful consideration of our manuscript, and in particular for such a careful attention to the results presented in Figure 1 of our initial submission. The reviewer is indeed right that there was a mistake in the figure. We have now redrawn the Figure so that it correctly reflects the negative correlation between IMD and fibre content. We have also double checked our analysis throughout, and edited our presentation to ensure it is correct.

=== 2. The key grocery elements of deprivation

There is very little discussion about the types of products that are more predictive of deprivation. I believe that the top predictors should be listed more exhaustively and discussed. I would also strongly suggest to perform three different predictions, one using just the nutrients (fibres, carbs..), one using just the item types (meat, sweets..), and one using all of them (the one that it is currently presented). It would be interesting to see which family of features is most predictive. Then, I would expect an expanded discussion or result section that contains some hypotheses (or even some speculations) of why some factors might be more predictive -- fibres is an easy example, but maybe there are more interesting associations to be discussed. This addition would add a considerable value to the contribution, in my opinion. If there are strange associations (i.e., not explained by intuition or previous work) it would be worth pointing them out.

We agree with the reviewer that this is a good suggestion, and we have carried out additional analysis to address this comment. We now present the results of three different models, as suggested by the reviewer. Overall, we find that the three different models perform relatively similarly, suggesting that the different categories all contain information related to deprivation levels. We have also added some discussion on the features which have the strongest predictive power, and why that may be. However, we also want to ensure that it is clear that our analysis does not provide any indication of causal relationships between grocery shopping habits and deprivation. We have tried to carefully word our discussion to ensure that no causal links can be implied by our discussion.

=== 3. The temporal aspect

I totally see why the authors decided to go for the full grocery dataset. Yet, a relevant piece of analysis that ties directly into their research question is how little data one can use to make a good IMD assessment from grocery data. At the moment, the authors are using one full year of data, but a sensitivity analysis that uses 1,2,3,6 months of data could reveal that smaller amounts of data could suffice for such an estimation.

This is a very appropriate comment, which we entirely agree should be included in our analysis. Therefore, we have extended our analysis to include a sequence of models which use increasingly more months of data, and we present this analysis in the results section, and in Figure 2D. Overall, we find that even just one month of data results in a model with a similar overall score to that of a model with one year of data.

=== Minor comments:

- Using decision trees with depth of 150 is definitely quite an overkill. This is also made quite clear by the analysis in the supplementary materials, which shows that the depth of the tree does not influence the results. I believe that a depth of 5 would do just as fine. I would strongly recommend to revise the presentation of the main results using trees of limited depth.

We thank the reviewer for their comment on this. We agree that a depth of 150 is not necessarily needed, so we have re-run our analysis setting the maximum depth of each tree in the forest to 30. It is also important to note that, whilst the maximum depth is set to 30, not every tree in the forest will be fully grown to that depth. We have edited this in the presentation of the manuscript as well.

- It is not necessary to repeat the analysis for IMD ranks and scores (in the supplementary materials). Those are completely symmetrical (so the results should not be "similar" but identical, just with a flipped sign). So just be very clear and explicit about which one you are using and about its meaning.

The reviewer raises a very relevant point here. In general, what the reviewer writes is correct. Our decision to carry out the analysis using the IMD scores was for two main reasons. Firstly, as described in the methodology reports of the IMD, the published ranks use slightly different (not rounded) versions compared to the published IMD scores, so minor differences may be present. Additionally, from a methodological perspective, our analysis wants to investigate whether performing a random forest regression on the ranks (which, technically, are ordered categories rather than numerical continuous variables) gives similar results to a regression on the continuous valued IMD scores. However, we agree with the reviewer that we don't need to report the results in such a level of detail as they are nearly the same. Therefore, we have revised our manuscript and supplementary information to only report that qualitatively similar results are found using the IMD scores.

- Which procedure did you use to implement the false discovery rate correction? It would be best to report it clearly (even just mentioning the name of the software package and function used)

We thank the reviewer for raising this. We have edited the Methods section to include an explicit reference to the implementation used for calculating the false discovery rate correction. In particular, we have used the openly available statsmodels Python package.

- There are no spaces before the in-text citations. For example, "available[36]" should be "available [36]". You can use `text~\cite{xyz}` to have a space and keep the citation close to the previous word.

Thank you for pointing out this formatting issue, which we have now corrected in the revised version of our manuscript.

Reviewer 2

This is a very interesting paper that looks at a very novel dataset in a novel setting. I very much enjoyed reading it. I think this is a promising paper, but needs revisions for a good publication.

We would like to thank the reviewer for their positive general feedback on our manuscript. We are pleased to hear that they enjoyed reading it and that they think it is a promising paper. We have extensively revised our manuscript based on the suggestions, and we believe that it has greatly improved from our initial submission.

Data and methods descriptions, and results need substantial improvements to clarify:

- **Page 4, Lines 50-51: Is there a full list of product categories and nutrients, along with how they are measured e.g. fractions in terms of items, grams, purchase value? Not clear from the text what the 188 features refer to. How come 19 of these features have a constant value? While there is a reference to another paper — I think the present paper should include a better and clearer definition here to make sure the reader can better understand the data used for this analysis.**

We agree that our previous presentation of the data could have been improved. We thank the reviewer for raising this issue. We have added some additional information about the grocery shopping data used, in terms of food categories and nutritional values, in order to provide more information about the data. We believe that the current presentation strikes a good balance between giving enough information about the data, whilst not reproducing exactly the presentation of the paper with the underlying data. It is important to highlight that including the definition of each variable would significantly extend this paper, and would effectively only replicate the thorough presentation of “Aiello LM, Quercia D, Schifanella R, Del Prete L. Tesco Grocery 1.0, a large-scale dataset of grocery purchases in London. Scientific data. 2020 Feb 18;7(1):1-1.”

- **If the IMD rankings are available for the whole UK, this means the rankings will take into account not only London LSOAs but also other LSOAs around the country. When being used for this analysis, do the rankings refer only to London, or UK. How might this affect results?**

We thank the reviewer for their comment on this. In our analysis, we only consider LSOAs in London. Before building any of our models, we re-rank the London LSOAs so that the ranks accurately reflect the fact that we are only considering London and not the whole UK. We have edited the text to make this clearer.

- Page 5, paragraph starting with Line 21 needs clarification - in its current format, it is very hard to understand what has been done for this analysis and why. Is there a reason

We thank the reviewer for this comment. We have edited the Methods section to improve clarity and ensure that all steps in our analysis are clear and reproducible by a reader.

Page 6, Lines 24-27 suggest that five-fold cross validation is done. Not clear how this is done with 70%-30% splits. This might be a typo, but then earlier Page 5, Lines 33-35 suggests that no cross validation is done, but only the 70-30% splits are done. This needs to be clarified. In the current setting, cross-validation is preferred. Results also need to reflect the differences in findings in different folds.

We agree with the reviewer that our presentation of this result was unclear and needed improving. The procedure that we use is to split the data with a 70%-30% split and we repeat this four times so that we consider four different random splits. We have edited the text to improve our presentation and clarify this aspect.

How is a single accuracy measure computed - on the full London data? Also Not sure where N=5,800 comes from. This seems higher than the number of LSOA's in London.

Thanks for raising this issue, which we agree needs clarifying. The accuracy measure is calculated as follows: we split the 4,833 LSOAs with a 70%-30% split, thus giving us 1,450 (30% of 4,833) LSOAs in the test set to use for predictions. We then repeat this four times, thus giving us a sample size of 5,800 (1,450 times four). We have edited our manuscript to clarify this.

Not clear what positive vs. negative correlation mean for Figure 2A (also in the text). The more the fibre, the better off the neighbourhood? Needs clarification what negative and positive means in the setting. Also, the paper will benefit from a discussion on whether these findings are intuitive or not. The more money(or items) as a fraction bought from Tesco, the worse off the neighbourhood? Do the authors find these results intuitive, and why?

We thank the reviewer for their comment on this important point. We have added further discussion on what a positive and negative correlation mean in this context. We have also edited our discussion to include further reflections on our findings.

Very hard to understand Figure 2D - the main text and caption are both confusing. Needs clarification. Are these all LSOAs in London, or only the 30%

We agree that the figure needed improving in clarity. We have now moved the figure to Figure 3A to be alongside the confusion matrix, as both figures refer to the accuracy of our analysis. We have also edited the presentation and caption of this to help the reader better understand the figure. More precisely, the figure only uses LSOAs in the test set (30% of the data), but over the four random repetitions discussed above. On the x-axis the figure reports the deciles derived from the predicted IMD ranks. The color of the bar refers to the true decile of the LSOAs. For instance, an LSOA whose true IMD decile is 10%, but which is incorrectly predicted to be in the 10%-20% decile by our model, would contribute to the count of the red bar in the 10%-20% group of the figure.

Do we know if/how Tesco's product ranges offered in different stores make a difference?

Whilst we agree that this would be very interesting to study, we unfortunately do not have this information available. We have added this as a limitation of our study and a possible avenue for further analysis should the data become available.

How are findings compare to existing literature on food choices by different income groups. There is a rich literature that focuses specifically on diets of different socioeconomic groups with a specific interest in poor neighbourhoods. The manuscript will benefit from clarifying the links to this literature base.

We thank the reviewer for their comment on this. We have edited our discussion section, and our manuscript throughout, to provide further links to the literature.

Not clear to me if Tesco data is also available for later years, and how it was accessed. It is publicly available - but I was not sure if the year 2015 was specifically selected to match the IMD data. If available from Tesco, data from the year 2019 will be key to understand if these datasets can be used for predictions. specifically, food preferences and contents change over time - we do not know how it will effect predictions. If this method was used in practice, one would need to train models from one year data and make predictions in following years. Not clear to me how the changing preferences and food content (from the supply side) will affect predictive capabilities. The paper should address such concerns. If 2019 data is available, this will be the ideal test case. Can a model trained on 2015 data make good predictions on 2019 data. As we already have 2019 IMD data, this can be tested.

We agree with the reviewer that using Tesco data from 2019 would be of great interest. Unfortunately, our analysis relies on data which was made publicly available by other researchers, and this data is only for the year 2015. Therefore we cannot use our model, trained on the 2015 data, to generate predictions for the 2019 IMD data. We agree that this would be the most natural next step in order to further test the ability of generating predictions in a practical setting.

It will be interesting to see how Tesco stores in addition to other retailers that cater to lower or higher income groups (Waitrose vs. Iceland) are distributed across space. Could that information be somehow used in relation to IMD. I would expect to observe a high correlation between existence of certain chains in certain LSOAs based on their target populations.

This is indeed a very interesting point and valuable suggestion, as we entirely expect that different retailers will appeal to different socioeconomic groups, such as those that the reviewer suggests. The analysis that the reviewer suggests would require us to be able to retrieve the location of all stores in London for each type of retailer, so that we can include it in our model. Whilst this may be possible with a combination of open data sources, such as OpenStreetMap and other publicly available data in London, we believe it goes beyond the scope of this study.

Last point on motivation. The authors motivate their work with an emphasis on prediction. Yet the train-test strategy (both over space, and over time) does not seem to be well-designed to fit the motivation. This needs clarification. Descriptive results on their own, are very interesting. The focus on predictive capabilities, however, needs a stronger argument and better justification for the train/test split strategies.

We thank the reviewer for their comment on this aspect of our work. We agree that an important focus of our work is on predictions. However, the overall goal is not to provide the most accurate predictions possible. Rather, we aim to show the existence of a link between deprivation and grocery shopping data, and that this link can be used to generate predictions with the available data, and, crucially, these predictions can be generated using a relatively simple model. We agree that our presentation of this aspect needed improving, and we have done two things to this purpose: first, we have significantly extended our analysis, by including an analysis over different months of data, as well as new models; secondly, we have edited the manuscript throughout to provide a clearer presentation of our analysis and results.